# Impact of chronic kidney disease on case ascertainment for hospitalised acute myocardial infarction: an English cohort study

Patrick Bidulka [1], Jemima Scott,[2,3] Dominic M Taylor,[2,3] Udaya Udayaraj,[4,5] Fergus Caskey,[2,3] Lucy Teece,[6] Michael Sweeting,[6] John Deanfield,[7,8] Mark de Belder,[7] Spiros Denaxas,[9,10] Clive Weston,[11] David Adlam,[12] Dorothea Nitsch[1]

PB and JS contributed equally. DA and DN contributed equally.

For numbered affiliations see end of article.

**Correspondence to**
Patrick Bidulka;
patrick.bidulka1@lshtm.ac.uk

## ABSTRACT

**Objectives** Acute myocardial infarction (AMI) case ascertainment improves for the UK general population using linked health data sets. Because care pathways for people with chronic kidney disease (CKD) change based on disease severity, AMI case ascertainment for these people may differ compared with the general population. We aimed to determine the association between CKD severity and AMI case ascertainment in two secondary care data sets, and the agreement in estimated glomerular filtration rate (eGFR) between the same data sets.

**Methods** We used a cohort study design. Primary care records for people with CKD or risk factors for CKD, identified using the National CKD Audit (2015–2017), were linked to the Myocardial Ischaemia National Audit Project (MINAP, 2007–2017) and Hospital Episode Statistics (HES, 2007–2017) secondary care registries. People with an AMI recorded in either MINAP, HES or both were included in the study cohort. CKD status was defined using eGFR, derived from the most recent serum creatinine value recorded in primary care. Moderate–severe CKD was defined as eGFR <60 mL/min/1.73 m$^2$, and mild CKD or at risk of CKD was defined as eGFR ≥60 mL/min/1.73 m$^2$ or eGFR missing. CKD stages were grouped as (1) At risk of CKD and Stages 1–2 (eGFR missing or ≥60 mL/min/1.73 m$^2$), (2) Stage 3a (eGFR 45–59 mL/min/1.73 m$^2$), (3) Stage 3b (eGFR 30–44 mL/min/1.73 m$^2$) and (4) Stages 4–5 (eGFR <30 mL/min/1.73 m$^2$).

**Results** We identified 6748 AMIs: 23% were recorded in both MINAP and HES, 66% in HES only and 11% in MINAP only. Compared with people at risk of CKD or with mild CKD, AMIs in people with moderate–severe CKD were more likely to be recorded in both MINAP and HES (42% vs 11%, respectively), or MINAP only (22% vs 5%), and less likely to be recorded in HES only (36% vs 84%). People with AMIs recorded in HES only or MINAP only had increased odds of death during hospitalisation compared with those recorded in both (adjusted OR 1.61, 95% CI 1.32 to 1.96 and OR 1.60, 95% CI 1.26 to 2.04, respectively). Agreement between eGFR at AMI admission (MINAP) and in primary care was poor (kappa (K) 0.42, SE 0.012).

## Strengths and limitations of this study

► Our study includes a large sample size of 6748 acute myocardial infarction (AMI) events.
► We have assessed the completeness of AMI hospitalisations recorded in two healthcare data sets widely used in observational research in England.
► We evaluated, for the first time, the validity of using serum creatinine recorded in secondary care at the time of an AMI to estimate pre-AMI chronic kidney disease (CKD) stage.
► Generalisability to the general population is limited as the National Chronic Kidney Disease Audit only included people with CKD and/or risk factors for CKD.

**Conclusions** AMI case ascertainment is incomplete in both MINAP and HES, and is associated with CKD severity.

## INTRODUCTION

Prognosis following acute myocardial infarction (AMI) has improved considerably over the past 50 years such that 85% of individuals now live longer than 1 year post-AMI.[1] Improved survival is the result of advances in AMI management, driven by evidence from large-scale randomised controlled trials (RCTs).[2–6] Of those admitted to hospital with AMI, 30%–40% have chronic kidney disease (CKD)[7]: a sustained reduction in kidney function associated with poor outcomes.[8 9] Among those with dialysis-dependent CKD only 40% will survive their first year post-AMI.[10] These inferior outcomes may result from higher prevalence of comorbidity,[2] calcific coronary artery disease[3] and the pro-inflammatory effects of uraemia.[4]

Most major RCTs investigating AMI interventions excluded patients with advanced CKD.[11] However, current AMI guidelines

from Europe and the USA apply the results of these RCTs to those with or without CKD.[5–7] Clinicians' unease with the dearth of evidence may explain diversion from these AMI guidelines when treating people with CKD.[2 10 11] In the absence of specific RCTs in CKD populations, well-conducted observational analyses can contribute significantly to our understanding and improved management of AMI.

In the UK, data on AMI treatment and outcomes is collected in unlinked, disease-specific registries or in broad registration databases. While there are known differences in the reliability and validity of AMI case ascertainment using these resources in the general population,[12] it is unclear to what extent these differences persist in people with underlying CKD. Multimorbidity and differences in admission and treatment pathways in people with CKD may influence AMI case recording. Reliably identifying which patients with AMI have CKD using AMI audit data is also difficult; previous studies used admission serum creatinine (SCr) as a proxy for pre-admission CKD stage.[13–15] This unvalidated method risks misclassifying people as having CKD because of the co-incidence of acute kidney injury (AKI) and AMI.[16]

In this study we linked records from the National Chronic Kidney Disease Audit (NCKDA) to the Myocardial Ischaemia National Audit Project (MINAP) and Hospital Episode Statistics (HES) to determine the reliability of these data sources to investigate cardiovascular disease comorbidity and outcomes in people with or at risk of CKD in England. Our objectives were to: (1) Compare case ascertainment of AMI hospitalisations in secondary care data sets (MINAP and HES); (2) determine if MINAP and/or HES case ascertainment defines populations of patients with CKD with different risks of death during and after AMI; and (3) compare CKD stage classification using admission SCr recorded in secondary care (MINAP) versus primary care (NCKDA).

## METHODS

### Data sources

Data from all sources were restricted to patients treated in England. People with or at risk of CKD were identified using primary care data from the NCKDA.[17 18] The NCKDA aimed to optimise the identification and management of people with CKD and/or risk factors for CKD in primary care, and included 10% of English General Practices (GP).[17 18] NCKDA data were collected between 2015 and 2016 in two main cross-sectional data extracts for people with either blood or urine laboratory results indicating CKD and/or risk factors for CKD (prevalent hypertension, diabetes mellitus, cardiovascular disease, connective tissue disorders, kidney stones, prostatic disease, family history of kidney disease, previous AKI and users of kidney-damaging medications such as lithium or calcineurin inhibitors).[17] People without an estimated glomerular filtration rate (eGFR) recorded in primary

care were included in the NCKDA only if they were at risk of CKD.

The population identified from NCKDA was linked with Office of National Statistics (ONS) data, January 1998 to September 2019, as well as secondary care data from HES Admitted Patient Care (APC) and MINAP, April 2007 to April 2017. HES APC includes hospital admission data for National Health Service-treated patients in England, including admission and discharge dates, and diagnoses recorded using International Classification of Diseases 10th Edition (ICD-10) codes.[19] MINAP is an ongoing AMI audit in England, Wales and Northern Ireland which was designed to optimise the care of patients with type one AMI by evaluating the patient pathway from hospital admission with AMI[20] to discharge. MINAP includes admission and discharge dates, treatments and comorbidities.[21]

### Study design

Cohort study.

### Study participants

We included people in the NCKDA registered with a GP in England, with one or more AMI hospitalisation recorded in HES or MINAP. People in each NCKDA extract must have been alive according to GP records at the time of that extract. We therefore included people with an AMI hospitalisation recorded in MINAP or HES only after the date of their GP's final NCKDA extract. People with an AMI hospitalisation that started prior to the extract date and ended after the extract date were added to the cohort, since they were at risk of death (n=183). In addition, people with an ONS death date indicating death during an AMI hospitalisation that occurred within 90 days prior to the NCKDA extract date were included (n=96), since they were likely misclassified as alive at the time of the extract because of delays in updating the death date in the GP systems. People with an ONS death date earlier than 90 days prior to the extract were excluded (n=4).

### Exposures

Our main exposure variable was moderate to severe CKD (eGFR <60 mL/min/1.73 m$^2$, CKD stages 3–5), defined using the most recent eGFR recorded in primary care (NCKDA data) prior to the AMI hospitalisation. People with no eGFR recorded in primary care or an eGFR ≥60 mL/min/1.73 m$^2$ were categorised as at risk of CKD or having mild CKD, respectively. We assumed people with no eGFR recorded in primary care did not have moderate to severe CKD since these people are much less likely to have CKD than those with eGFR recorded.[22] eGFR was calculated using primary care SCr measures and the revised Modification of Diet in Renal Disease (MDRD) equation.[23 24]

Our secondary exposure was CKD stage, defined by the Kidney Disease Improving Global Outcomes CKD staging, based on a single eGFR record without the requirement for two measures 3 months apart.[25] We combined some CKD stages due to low numbers of AMI cases: (1) At

risk of CKD and Stages 1–2 (eGFR missing or ≥60 mL/min/1.73 m$^2$), (2) Stage 3a (eGFR 45–59 mL/min/1.73 m$^2$), (3) Stage 3b (eGFR 30–44 mL/min/1.73 m$^2$) and (4) Stages 4–5 (eGFR <30 mL/min/1.73 m$^2$).

We used the latest eGFR recorded prior to the AMI hospitalisation to categorise people with a history of kidney transplant into the primary and secondary exposure groups. We categorised people with a history of dialysis prior to the AMI hospitalisation as moderate to severe CKD for the main exposure and CKD stages 4–5 for the secondary exposure, even if the latest eGFR did not agree.

As the use of a single SCr test at the time of AMI hospitalisation to determine CKD stage has not previously been validated, we have used the term 'eGFR stage' in place of CKD stage to refer to the eGFR level calculated from this test.

## Outcomes
### Primary outcome
The primary outcome was AMI case ascertainment, defined as the data set(s) in which the AMI hospitalisation was recorded. We defined an AMI as being recorded in both HES and MINAP if an AMI hospitalisation in HES was within 30 days of an AMI hospitalisation in MINAP. Where multiple HES AMI hospitalisations fell within 30 days of a MINAP AMI hospitalisation, the HES AMI hospitalisation closest in time to the MINAP AMI admission was selected as the single matched event. AMI hospitalisations without a match were categorised as HES or MINAP only. Study participants could contribute multiple AMI hospitalisations.

We defined an AMI in HES data using ICD-10 codes I.21, I.22 or I.23 in the primary admission diagnosis field (first diagnostic position in the first episode of an admission).[26] We categorised AMI subtype (ST-segment elevation myocardial infarction (STEMI) and non-ST-segment elevation myocardial infarction (NSTEMI)) using the UK Biobank coding definitions[27] (online supplemental table 1). We defined AMI in MINAP using a previously developed algorithm which uses the discharge diagnosis, ECG results and the presence of elevated cardiac markers to identify acute coronary syndrome events and subtypes (online supplemental table 2). We excluded MINAP hospitalisations classified as unstable angina or other from the analysis.[12]

### Secondary outcomes
We investigated in-hospital mortality during each person's first AMI hospitalisation within the study period. In those who survived and were discharged from their first AMI hospitalisation, we also investigated post-discharge mortality using the ONS death date (up to 15 September 2019). Variables in HES, MINAP and ONS used to define death are described in online supplemental table 3. People were considered to have died during AMI hospitalisation if any of these variables indicated in-hospital death, or the ONS death date fell on or between the admission and discharge dates. We used the earliest of the HES or MINAP admission dates and the latest of the HES or MINAP discharge dates to define these dates for AMI hospitalisations recorded in both data sets.

We investigated the agreement between CKD stage derived from the most recent primary care SCr test (NCKDA data) and eGFR stage derived from the secondary care SCr test conducted within 24 hours of AMI hospitalisation (MINAP data). We used the same methods to determine eGFR stage in MINAP data as we did for NCKDA data.

## Covariates
We described age at AMI admission (mean and SD as well as age category in years: 18–49, 50–64, 65–79, 80+), sex, ethnicity (white or other), index of multiple deprivation quintiles (IMD, as a proxy for socioeconomic status) and relevant comorbidities including angina, cerebrovascular disease, chronic obstruction pulmonary disease (COPD), diabetes mellitus (type 1 and 2), heart failure, hypertension, previous myocardial infarction and peripheral vascular disease. We also described dialysis and transplant status, and smoking status. Data sources for each key covariate are described in online supplemental table 4.

## Data analysis
### Objective 1—AMI case ascertainment
We summarised key covariates by CKD status. We used Venn diagrams to describe AMI case ascertainment overall and stratified by CKD status (at risk of CKD or mild CKD vs moderate to severe CKD). We used multinomial, multivariable logistic regression to quantify the association between CKD stage and AMI case ascertainment. We used the 'HES and MINAP' category as the base outcome and reported crude and adjusted relative risk ratios (RRR) and 95% CIs, using 'At risk of CKD and Stages 1–2' as the reference exposure category. We adjusted for sex, age category, ethnicity, IMD quintile, previous AMI, heart failure, COPD, diabetes mellitus and clustering by participant (using cluster-robust standard errors (SEs)). We used a complete case analysis since we could not assume that missing values for ethnicity were missing at random. In a secondary analysis, we stratified these regressions by AMI subtype (STEMI and NSTEMI).

### Objective 2—risk of death
We used multivariable logistic regression to calculate the odds of death in hospital during each person's first AMI hospitalisation in people with AMI recorded in MINAP only or HES only, relative to MINAP and HES. After confirming the proportional hazards assumption with a Schoenfeld Residuals test on the full multivariable model (p=0.35), we used multivariable Cox regression to estimate HRs for death during total follow-up in those who survived their first AMI hospitalisation with AMI recorded in MINAP only or HES only, relative to MINAP and HES.

### Objective 3—agreement between eGFR in primary and secondary care

Finally, to assess the validity of using MINAP-recorded eGFR at AMI admission as a proxy for pre-admission CKD status, we compared eGFR and its corresponding eGFR stage within 24 hours of AMI admission (MINAP data) to the most recent eGFR and its corresponding CKD stage in primary care (NCKDA data). In this analysis, we excluded people with eGFR measures greater than $120\,mL/min/1.73\ m^2$ in either NCKDA or MINAP as these are unlikely to be true values. We drew a Bland-Altman plot to describe differences in the distribution of eGFR measures in primary and secondary care[28] and calculated the per cent agreement and kappa agreement statistics between CKD and eGFR stage derived using primary and secondary care eGFRs, respectively. Secondary analyses re-calculated agreements and kappa statistics restricting to people with stages 3a–5 (eGFR $<60\,mL/min/1.73\ m^2$) in primary care and grouped by time between the most recent primary care eGFR measure and the AMI hospitalisation (0–5, 6–11, 12–23, 24–36 month gaps).

### Sensitivity analyses

We repeated the main analyses for AMI events occurring prior to the study start date (the latest NCKDA extract). People who experienced AMI hospitalisation before the study start were survivors, since only people alive at the time of the NCKDA were included in the study.

In addition, we re-drew the Venn diagrams after including HES AMI hospitalisations recorded in both the first and second diagnostic positions of the first episode to include AMIs recorded as co-primary diagnoses.[26] We also repeated the matching process between MINAP and HES AMI hospitalisations after combining all HES AMIs within 30 days of each other into a single HES hospitalisation, since it is likely that some AMI events in our data set have multiple HES hospitalisations recorded if, for example, a patient is transferred between hospitals for treatment. Furthermore, we repeated our multivariable analyses after excluding people with a history of dialysis. Finally, to understand why people may have an AMI hospitalisation recorded in MINAP but not HES, we searched for non-AMI HES hospitalisations within 30 days of the AMI recorded in MINAP and described the ICD-10 diagnoses in the first episode of the first diagnostic position.

### Missing data

We did a complete case analysis when building our multivariable models. People with missing ethnicity (~1%) and IMD data (<1%) were excluded prior to building our unadjusted, partially adjusted and adjusted multinomial models.

We used discharge dates to help re-categorise people who were in-hospital at the time of the NCKDA extract into the cohort, as well as to determine death in hospital and the start of follow-up in those who survived their first AMI hospitalisation. Discharge date was missing in 19% and 1% of the MINAP and HES data sets, respectively. We

assumed these dates were missing at random and used the median length of admissions in those without missing admission and discharge dates (5 and 4 days in MINAP and HES, respectively) to impute the missing discharge dates.

### Patient and public involvement

The Kidney Care UK patient organisation (https://www.kidneycareuk.org/) supported the research questions, grant applications and the related record linkage application for section 251 permissions critical to the development of the NCKDA. Patient members of the UK Renal Registry Patient Council (https://renal.org/patients/patient-council) reviewed the study results. Their feedback supported a further planned record linkage of renal and cardiac data to look at patient outcomes.

## RESULTS

### Study population and baseline characteristics

From 1 702 345 people in England included in the NCKDA, we identified 6042 (0.4%) people with or at risk of CKD who experienced 6748 AMIs between the final NCKDA extract and 1 April 2017 (online supplemental figure 1). Baseline characteristics stratified by CKD stage are described in table 1. People with moderate to severe CKD accounted for 38% of AMI hospitalisations (2,575). Average age at the time of AMI was 73 years (SD 13). People with moderate to severe CKD were older on average than people with mild CKD or at risk of CKD. Most people were white (92%) and men (61%). The most prevalent comorbidities were hypertension (61%) and diabetes mellitus (35%).

### AMI recording in HES and MINAP

Overall, 23% of AMI hospitalisations were captured by both MINAP and HES data sets (1552 AMI hospitalisations) (figure 1). There was no substantial change in AMI case ascertainment over time (online supplemental figure 2). In people with moderate to severe CKD, 42% of all AMI hospitalisations were captured by both MINAP and HES (1092 AMI hospitalisations). In people with mild CKD or at risk of CKD, 11% of all AMI hospitalisations were captured by MINAP and HES (460 AMI hospitalisations) (figure 1).

### Relative association between CKD stage and AMI recording

Crude and adjusted RRRs and 95% CIs describing the association between CKD stage and AMI case ascertainment are presented in table 2. After adjusting for key covariates, we observed weak evidence of an increased likelihood of AMI recorded in MINAP only, compared with MINAP and HES, in people with CKD stages 4–5 versus the at risk of CKD/stages 1–2 group (RRR 1.34, 95% CI 0.97 to 1.85). Furthermore, compared with the at-risk of CKD/stages 1–2 group, people with CKD stages 3a, 3b and 4–5 were less likely to have an AMI hospitalisation recorded in HES only versus MINAP

**Table 1** Baseline characteristics by CKD stage for all AMI events captured after the study start. n (column %) unless specified otherwise.

| CKD status (main exposure) | At risk of CKD or mild CKD | Moderate to severe CKD | | | |
|---|---|---|---|---|---|
| CKD stage (secondary exposure) | At risk of CKD and 1–2 | 3a | 3b | 4–5 | Total |
| Unique individuals | 3751 | 1210 | 732 | 349 | 6042 |
| Total number of AMI events, N | 4173 | 1353 | 825 | 397 | 6748 |
| Age at AMI event, years, mean (SD) | 70 (13) | 79 (10) | 82 (9) | 79 (12) | 73 (13) |
| Age category at AMI event, years | | | | | |
| 18–50 | 299 (7) | 20 (1) | <5 | 10 (3) | 330 (5) |
| 50–64 | 1163 (28) | 91 (7) | 44 (5) | 38 (10) | 1336 (20) |
| 65–79 | 1655 (40) | 531 (39) | 251 (30) | 122 (31) | 2559 (38) |
| 80+ | 1056 (25) | 711 (53) | 529 (64) | 227 (57) | 2523 (37) |
| Female | 1430 (34) | 638 (47) | 416 (50) | 168 (42) | 2652 (39) |
| Ethnicity | | | | | |
| White | 3807 (91) | 1263 (93) | 773 (94) | 361 (91) | 6204 (92) |
| Other | 323 (8) | 73 (5) | 44 (5) | 34 (9) | 474 (7) |
| Missing | 43 (1) | 17 (1) | 8 (1) | <5 | 70 (1) |
| IMD quintile | | | | | |
| 1 (least deprived) | 732 (18) | 252 (19) | 152 (18) | 68 (17) | 1204 (18) |
| 2 | 843 (20) | 306 (23) | 176 (21) | 85 (21) | 1410 (21) |
| 3 | 934 (22) | 331 (24) | 193 (23) | 79 (20) | 1537 (23) |
| 4 | 951 (23) | 255 (19) | 185 (22) | 96 (24) | 1487 (22) |
| 5 (most deprived) | 690 (17) | 206 (15) | 114 (14) | 69 (17) | 1079 (16) |
| Missing | 23 (1) | <5 | 5 (1) | 0 (0) | 31 (0) |
| Dialysis in primary care | | | | | |
| Peritoneal dialysis | 0 (0) | 0 (0) | 0 (0) | 15 (4) | 15 (0) |
| Haemodialysis | 0 (0) | 0 (0) | 0 (0) | 24 (6) | 24 (0) |
| Renal dialysis, unspecified | 0 (0) | 0 (0) | 0 (0) | 7 (2) | 7 (0) |
| Kidney transplant | 5 (0) | 0 (0) | <5 | 17 (4) | 26 (0) |
| Comorbidities | | | | | |
| Angina | 959 (23) | 399 (29) | 275 (33) | 155 (39) | 1788 (26) |
| Cerebrovascular disease | 390 (9) | 178 (13) | 139 (17) | 81 (20) | 788 (12) |
| COPD | 514 (12) | 209 (15) | 168 (20) | 58 (15) | 949 (14) |
| Diabetes mellitus | 1293 (31) | 465 (34) | 356 (43) | 233 (59) | 2347 (35) |
| Heart failure | 400 (10) | 234 (17) | 211 (26) | 123 (31) | 968 (14) |
| Hypertension | 2333 (56) | 884 (65) | 583 (71) | 322 (81) | 4122 (61) |
| Myocardial infarction | 1050 (25) | 430 (32) | 274 (33) | 163 (41) | 1917 (28) |
| Peripheral vascular disease | 229 (5) | 108 (8) | 74 (9) | 47 (12) | 458 (7) |
| Smoking status | | | | | |
| Non-smoker | 1953 (47) | 566 (42) | 306 (37) | 151 (38) | 2976 (44) |
| Ever-smoker | 2018 (48) | 530 (39) | 318 (39) | 140 (35) | 3006 (45) |
| Missing | 202 (5) | 257 (19) | 201 (24) | 106 (27) | 766 (11) |

AMI, acute myocardial infarction; CKD, chronic kidney disease; COPD, chronic obstructive pulmonary disease; IMD, Index of Multiple Deprivation.;

and HES. We did not observe any differences in the likelihood of recording of AMI hospitalisation when stratifying by AMI subtype (online supplemental table 5).

**Mortality during AMI hospitalisation and post-discharge**

Of those with a first AMI recorded in both HES and MINAP, 209 people (15%) died during the AMI hospitalisation, compared with 151 (23%) with a first AMI

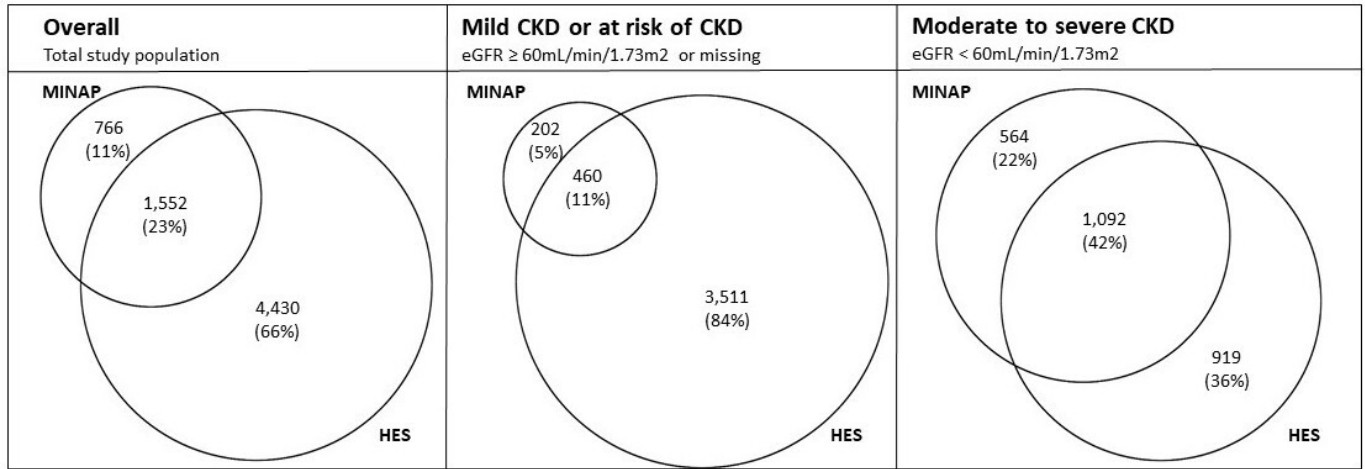

**Figure 1** Venn diagrams illustrating acute myocardial infarction (AMI) recording in MINAP and HES secondary care data sets. Venn diagrams presented overall, and stratified by CKD status (at-risk of or mild CKD, eGFR ≥60 mL/min/1.73 m² or moderate to severe CKD, eGFR <60 mL/min/1.73 m²). Circle areas are proportional to the number of AMI events in each data set. CKD, chronic kidney disease, eGFR, estimated glomerular filtration rate, HES, hospital episode statistics, MINAP, Myocardial Ischaemia National Audit Project.

recorded in MINAP only and 579 (15%) recorded in HES only (table 3). After adjusting for key covariates, people with AMI recorded in MINAP only and HES only had increased odds of in-hospital death compared with people with AMI recorded in both MINAP and HES (OR 1.60, 95% CI 1.26 to 2.04 and OR 1.61, 95% CI 1.32 to 1.96, respectively).

Mean follow-up among people who survived a first AMI hospitalisation was 2.4 years. The rate of death per 100 person-years during complete follow-up was 18.0 (95% CI 16.4 to 19.7) for AMI recorded in MINAP and HES,

23.3 (95% CI 20.6 to 26.5) for AMI recorded in MINAP only and 10.3 (95% CI 9.61 to 11.0) for AMI recorded in HES only (table 3). After adjusting for key covariates, there was no evidence of a difference in death during follow-up based on which data set(s) captured the first AMI hospitalisation.

### Agreement between eGFR derived from secondary care versus primary care data

Of the AMI hospitalisations recorded in MINAP, 2240 (97%) had SCr recorded within 24 hours of AMI admission

**Table 2** Multinomial logistic regression comparing the RRR of AMI recording across HES and MINAP according to CKD stage. The comparator outcome is people with AMI recorded in both HES and MINAP databases.

| AMI recording (outcome, compared with people with AMI recorded in MINAP and HES) | CKD stage (exposure) | Number of AMI admissions, n= | Unadjusted* RR (95% CI) | Partially adjusted† RRR (95% CI) | Adjusted‡ RRR (95% CI) |
|---|---|---|---|---|---|
| MINAP only (N=742) | At risk of CKD/stages 1–2 | 196 | 1 | 1 | 1 |
| | Stage 3a | 245 | 1.07 (0.85 to 1.34) | 0.98 (0.77 to 1.25) | 0.98 (0.77 to 1.25) |
| | Stage 3b | 197 | 1.17 (0.92 to 1.49) | 1.04 (0.80 to 1.36) | 1.03 (0.79 to 1.34) |
| | Stages 4–5 | 104 | 1.50 (1.11 to 2.03) | 1.38 (1.01 to 1.90) | 1.34 (0.97 to 1.85) |
| HES only (N=4367) | At risk of CKD/stages 1–2 | 3456 | 1 | 1 | 1 |
| | Stage 3a | 557 | 0.14 (0.12 to 0.16) | 0.14 (0.12 to 0.17) | 0.14 (0.12 to 0.16) |
| | Stage 3b | 224 | 0.08 (0.06 to 0.09) | 0.08 (0.07 to 0.10) | 0.08 (0.06 to 0.10) |
| | Stages 4–5 | 130 | 0.11 (0.08 to 0.14) | 0.11 (0.09 to 0.15) | 0.12 (0.09 to 0.16) |

*Complete cases for adjusted model.
†Adjusted for sex, age at AMI admission, ethnicity (white, other), IMD quintile, clustering by participant.
‡Additionally adjusted for previous AMI, heart failure, COPD, diabetes mellitus.
AMI, acute myocardial infarction; CKD, chronic kidney disease; COPD, chronic obstructive pulmonary disease; HES, Hospital Episode Statistics; IMD, Index of Multiple Deprivation; MINAP, Myocardial Ischaemia National Audit Project; RRR, relative risk ratios.

**Table 3** Death during and after first AMI hospitalisation in total study population at risk of or with CKD

| Death during first AMI hospitalisation (N=5919)* | Number who died, n (%) | – | Unadjusted OR (95% CI) | Adjusted† OR (95% CI) |
|---|---|---|---|---|
| MINAP and HES | 209 (15) | – | 1 | 1 |
| MINAP only | 151 (23) | – | 1.67 (1.32 to 2.11) | 1.60 (1.26 to 2.04) |
| HES only | 579 (15) | – | 0.98 (0.82 to 1.16) | 1.61 (1.32 to 1.96) |

| Death during complete follow-up in those who survive first AMI hospitalisation (N=5009)* | Number who died during follow-up, n | Rate per 100 person-years (95% CI) | Unadjusted HR (95% CI) | Adjusted† HR (95% CI) |
|---|---|---|---|---|
| MINAP and HES | 456 | 18.0 (16.4 to 19.7) | 1 | 1 |
| MINAP only | 237 | 23.3 (20.6 to 26.5) | 1.27 (1.08 to 1.48) | 1.12 (0.96 to 1.31) |
| HES only | 847 | 10.3 (9.61 to 11.0) | 0.59 (0.53 to 0.67) | 1.07 (0.94 to 1.21) |

*Complete cases for adjusted model.
†Adjusted for sex, age at AMI admission, ethnicity (white, other), IMD quintile, previous AMI, heart failure, COPD, diabetes mellitus.
AMI, acute myocardial infarction; CKD, chronic kidney disease; COPD, chronic obstructive pulmonary disease; HES, Hospital Episode Statistics; IMD, Index of Multiple Deprivation; MINAP, Myocardial Ischaemia National Audit Project; OR, odds ratio.

(online supplemental table 6). Median eGFR at time of admission was $47.6 \, \text{mL/min/1.73 m}^2$ (IQR 33.5–61.6). The Bland-Altman plot comparing the primary care eGFR and the secondary care eGFR indicated a negligible mean difference but wide variation (mean difference $3.35 \, \text{mL/min/1.73 m}^2$, 95% CI –23.4 to 30.1) (online supplemental figure 3).

The per cent agreements and kappa statistics between eGFR stage derived from MINAP eGFR at AMI admission and the CKD stage derived from NCKDA using primary care data are shown in table 4. Overall, there was 57.2% agreement in staging (kappa statistic (K) 0.42 (SE 0.012)). When restricting to people with NCKDA-derived CKD stages 3–5, the % agreement and K indicated worse agreement (table 4). However, when looking at agreement in categorising people as having moderate to severe CKD (stages 3–5) versus mild CKD (stages 1–2), agreement improved (82.1% agreement, K 0.55 (SE 0.021)).

When stratifying by months between the primary and secondary care eGFR measures, we observed the best agreement in staging within a 0–5 month gap between the primary and secondary care eGFR measures: 61.0%, K 0.48 (SE 0.03) (table 4). Agreement was worse when the time between eGFR measures increased.

### Sensitivity analyses

AMI case ascertainment in MINAP and HES was similar in AMI hospitalisations recorded prior to the study start (sensitivity analysis) compared with after the study start (main analysis) (online supplemental tables 7-8, figure 4). There were also no major differences in agreement between CKD staging derived in primary versus secondary care when investigating AMI hospitalisations prior to the study start (online supplemental table 9).

After expanding the AMI definition in HES to include any hospitalisations with AMI coded in the second diagnostic position as well as the first, the proportion of AMI hospitalisations captured in both HES and MINAP

decreased slightly (online supplemental figure 5). After combining HES AMI admissions within 30 days of each other for the same person, we observed a 1% increase in the proportion of AMI hospitalisations recorded in both MINAP and HES (online supplemental figure 5). Results were also similar when excluding people with a history of dialysis (online supplemental tables 10-11).

**Table 4** Agreement between primary care-derived CKD stage (NCKDA) and secondary care-derived eGFR stage (MINAP)

| | % agreement | Kappa statistic (SE) |
|---|---|---|
| Overall* | 57.2 | 0.42 (0.012) |
| CKD stages 3a, 3b, 4–5† | 53.2 | 0.34 (0.015) |
| CKD stages 1–2, 3a–5‡ | 82.1 | 0.55 (0.021) |
| **Overall,* by time from NCKDA SCr test (primary care) to MINAP SCr test (at AMI secondary care admission)** | | |
| 0–5 months | 61.0 | 0.48 (0.03) |
| 6–11 months | 56.7 | 0.42 (0.02) |
| 12–23 months | 55.9 | 0.40 (0.02) |
| 24–36 months | 56.8 | 0.41 (0.04) |

*Overall agreement when grouping as (1) Stages 1–2 (eGFR 60–120 mL/min/1.73 m$^2$), (2) Stage 3a (eGFR 45–59 mL/min/1.73 m$^2$), (3) Stage 3b (eGFR 30–44 mL/min/1.73 m$^2$) and (4) Stages 4–5 (eGFR 0–30 mL/min/1.73 m$^2$).
†Agreement when restricting to people with CKD stages 3a–5, grouped as (1) Stage 3a (eGFR 45–59 mL/min/1.73 m$^2$), (2) Stage 3b (eGFR 30–44 mL/min/1.73 m$^2$) and (3) Stages 4–5 (eGFR 0–30 mL/min/1.73 m$^2$).
‡Agreement when grouping as (1) Stages 1–2 (eGFR 60–120 mL/min/1.73 m$^2$) and (2) Stages 3a–5 (eGFR 0–59 mL/min/1.73 m$^2$).
AMI, acute myocardial infarction; CKD, chronic kidney disease; eGFR, estimated glomerular filtration rate; MINAP, Myocardial Ischaemia National Audit Project; NCKDA, National Chronic Kidney Disease Audit; SCr, serum creatinine; SE, Standard error.

Finally, the 10 most common diagnoses in HES matching with the MINAP only AMI hospitalisations from the main analysis are shown in online supplemental table 12. Eighty-eight per cent of unmatched MINAP AMIs had a non-AMI HES hospitalisation within 30 days. These were mainly CVD or respiratory infection-related ICD-10 diagnoses.

## DISCUSSION

We compared recording of AMI hospitalisations for people with CKD between two large secondary healthcare data sets in England. In a cohort of 6042 people, we found that both HES and MINAP missed a significant proportion of AMI hospitalisations. CKD stage influenced likelihood of AMI recording by data set: AMI hospitalisations in people with moderate to severe CKD were more likely to be recorded in MINAP compared with people at risk of CKD or with mild CKD. We found an association between AMI hospitalisation recording by data set and in-hospital mortality. There was marked variation between eGFR at AMI admission and preceding eGFR measurements in primary care, but no obvious systematic bias in terms of over/underestimation of eGFR at AMI admission.

Our results agree with previous research demonstrating incomplete capture of AMI events by individual healthcare data sets in the overall English population and extend them to a population with CKD. Herrett et al[12] showed 46% agreement when restricting to MINAP and HES recorded AMI hospitalisations, which is close to the 42% agreement we found in people with moderate to severe CKD. A smaller single-centre study by Torabi et al[29] found 32% agreement between MINAP and the hospital information department (responsible for HES coding).

In contrast to both studies, we found significantly worse agreement in case ascertainment for AMI hospitalisations between MINAP and HES for people at risk of CKD or with mild CKD. Torabi et al[29] collected data on renal function but found likelihood of AMI case recording in MINAP to reduce with advancing CKD stage. Differences in results between this and our study could be ascribed to changes in management of patients and/or event recording over time, differences in the populations studied and local practice in the single centre analysed by Torabi et al[29]

The high prevalence of CKD risk factors in people at risk of or with mild CKD could put them at greater risk of type 2 AMI; a mismatch of myocardial oxygen supply and demand in the absence of the 'classical' coronary artery plaque rupture with thrombosis reflective of type 1 AMI.[20] People with type 2 AMI are typically older, with a greater burden of comorbidities than those with type 1 AMI, and have poor outcomes.[30] HES is likely to include more type 2 AMI than MINAP as clinical coders for the latter are asked to select type 1 AMI only.

People with AMI recorded in both MINAP and HES had lower in-hospital mortality compared with those with AMI recorded in either MINAP or HES only. Our findings agree with Herrett et al[12]; patients with AMI recorded in only one source had a higher mortality than those with events recorded in more than one source. Higher in-hospital mortality in the MINAP only cases is likely to reflect the referral of severe and complex AMI cases to cardiology, including a higher STEMI to NSTEMI ratio.

Across all levels of eGFR, we found significant variation between eGFR stage derived from SCr taken within 24 hours of AMI admission (recorded in MINAP) and that derived from SCr in primary care, which is in line with reported variability of eGFR in validation studies.[23 24] As expected with known limitations of using MDRD eGFR to estimate kidney function for GFRs above 60 mL/min/1.73 m$^2$, binary classification between individuals with CKD stages 3–5 and those with stages 1–2 is more reliable than classification by CKD stage. These findings suggest that although previous research[13–15] using SCr at AMI admission recorded in MINAP as a proxy for baseline CKD stage may result in misclassification, it is unlikely to have resulted in a systematic bias in either overestimation or underestimation of CKD stage, despite our initial hypothesis that there would be systematic underestimation of kidney function due to the substantially increased risk of AKI during an AMI hospitalisation.[31] Differences between SCr recorded in primary care and SCr recorded in MINAP may reflect progression of CKD, differential use of medication that affects the renin–angiotensin–aldosterone system, AKI at the time of serum sampling (although changes in SCr are unlikely to show within 24 hours of AMI onset), or variation around the mean.

### Limitations

The NCKDA only included people with CKD and/or risk factors for CKD; therefore, we cannot generalise our results to people without risk factors for CKD. We may have incorrectly misclassified people who have no documented tests for CKD in primary care as having risk factors for CKD only; however, previous work has shown this group of people are much less likely to have CKD than those who do have CKD tests recorded in primary care.[22] Furthermore, we included people with at least one reduced kidney function test as potentially having CKD since not every patient undergoes regular CKD testing in our routine clinical data sets. Defining CKD using one eGFR measurement will have led to some misclassification. However, as people with CKD have very high cardiovascular risk and because of the infrequent SCr measurement in primary care, applying the chronicity criterion would have led to a selected cohort of people who did not develop a myocardial infarction until the second measurement had been done. Our results are likely impacted by residual confounding, since we were limited in the number of relevant comorbidities we could include in our multivariable models. Finally, AMI misclassification in HES data may have occurred due to the structure and level of detail available in this data set. For example, we may have missed AMI cases by including only those recorded in the first diagnostic position of the first episode of an HES admission; however, our sensitivity

analysis which included AMI hospitalisations recorded in the first or second diagnostic position showed similar results. In addition, unlike MINAP data, HES data do not include ECG results and troponin levels, which we could have used to reduce potential misclassification. Inclusion of the first diagnostic position of later episodes was undertaken in a similar study investigating AMI case ascertainment in people with malignancy, with little improvement in agreement between data sets.[32]

## Future research

This study demonstrates how AMI case ascertainment in England can be improved by using linked healthcare data sets. Further research investigating cardiovascular and kidney disease incidence, prevalence and outcomes should follow this approach. Other countries with similarly rich, yet fragmented healthcare data sets would benefit from applying similar methods to evaluate the validity and completeness of cardiovascular and kidney disease capture in similar data. Optimising data quality in healthcare data sets and simplifying the process of data linkage would facilitate high-quality observational research to inform the design of future RCTs and provide estimated treatment effects where RCT data are lacking.

## CONCLUSION

The use of linked healthcare data sets should be prioritised in observational research investigating multimorbidity.

**Author affiliations**
[1]Department of Non-Communicable Disease Epidemiology, London School of Hygiene & Tropical Medicine, London, UK
[2]Population Health Sciences, University of Bristol, Bristol, UK
[3]Richard Bright Renal Service, North Bristol NHS Trust, Southmead Hospital, Bristol, UK
[4]Oxford Kidney Unit, Churchill Hospital, Oxford, UK
[5]Nuffield Department of Medicine, University of Oxford, Oxford, UK
[6]Biostatistics Research Group, Department of Health Sciences, University of Leicester, UK
[7]National Institute for Cardiovascular Outcomes Research (NICOR), Barts Health NHS Trust, London, UK
[8]Institute of Cardiovascular Sciences, University College London, London, UK
[9]Institute of Health Informatics, Faculty of Population Health Sciences, University College London, London, UK
[10]Health Data Research UK, London, UK
[11]Glangwili General Hospital, Carmarthen, UK
[12]Department of Cardiovascular Sciences, University of Leicester and NIHR Leicester Biomedical Research Centre, Leicester, UK

**Acknowledgements** MINAP is commissioned, as part of the 'National Cardiac Audit Project' by the Healthcare Quality Improvement Partnership. This work uses data that has been provided by patients and collected by the NHS as part of their care and support. We acknowledge and thank UK cardiovascular, renal and primary care physicians as well as hospital and community audit and coding teams whose diligent data collection has provided the data for these analyses. In particular, we would like to thank James Chal, Anil Gunesh, Andrew Harrison, and Akosua Donkor from NICOR; and Kathryn Griffith, Matthew Harker, Yvonne Silove, Tasneem Hoosain, Nick Wilson, Ronnie Moodley, Richard Fluck, Chris Gush, David Wheeler, Liam Smeeth, Ron Cullen, Andy Syme, Richard Gunn, Paul Wright, Hugh Gallagher, Sion Edwards, Fiona Loud, Nick Palmer, Richard Fluck, Anita Sharma, Kate Cheema, Andy Syme, Sally Hull, Ben Caplin, Lois Kim, and Faye Cleary from the NCKDA steering, clinical reference and statistical groups.

**Contributors** JS, DT, DN, DA, FC and UU conceived the study. JS, DT, DN, DA, FC, UU and PB designed the study. PB and DN had access to and analysed the data. PB and JS drafted the manuscript. PB, JS, DT, UU, FC, LT, MS, JD, MdB, SD, CW, DA and DN (all authors) critically appraised the results and edited the manuscript. All authors approved the final version of the manuscript. The lead authors confirm all authors meet ICJME authorship criteria, and no one who meets ICJME criteria have been excluded. PB and DN are the guarantors of this study, and accept full responsibility for the work and conduct of the study, had access to the data, and controlled the decision to publish.

**Funding** This work was supported by Kidney Research UK (grant number IN_008_20180304) and the Health Foundation (grant number 1725841). JS is a Doctoral Research Fellow funded by the NIHR (NIHR 300906). DA is funded by a joint research grant from the British Heart Foundation (SP/16/5/32415) and Cancer Research UK (C53325/A21134).

**Competing interests** All authors have completed an ICJME form. PB, JS, DT, FC, LT, MdB and SD have nothing to declare. UU declares a grant from the Health Foundation to undertake quality improvement unrelated to this work. MS declares funding from Cancer Research UK (C53325/A21134) and British Heart Foundation (SP/16/5/32415) for research activities related to this work looking at acute myocardial infarction ascertainment in a cancer population. JS declares Doctoral research funding from the NIHR for related research looking at acute myocardial infarction care for people with chronic kidney disease. JD declares grants from British Heart Foundation paid to his institution unrelated to this work. JD also declares consulting fees from Novo Nordisk, and honoraria from Amgen, Boehringer Ingelheim, Merck, Pfizer, Aegerion, Novartis, Sanofi, Takeda, Novo Nordisk and Bayer, unrelated to this work. JD is also member of a study steering committee with Novo Nordisk. CW declares he is clinical lead of the Myocardial Ischaemia National Audit Project. DA reports research funding and in-kind support from AstraZeneca for unrelated research and educational funding from Abbott Vascular to support a clinical research fellow doing unrelated research. DA has conducted consultancy for General Electric to support general research funds. DN is the UK Kidney Association Director of Informatics Research. DN is also on the steering group for two GlaxoSmithKline funded studies that investigate kidney function in children and adults in sub-Saharan Africa.

**Patient and public involvement** Patients and/or the public were involved in the design, or conduct, or reporting, or dissemination plans of this research. Refer to the Methods section for further details.

**Patient consent for publication** Not applicable.

**Ethics approval** This study was approved by the London School of Hygiene & Tropical Medicine LEO ethics committee (ID 16988) and the National Chronic Kidney Disease Audit (NCKDA) Steering Committee. Audits used in this study are covered by Section 251 approvals (NHS Act 2006) which allows data to be collected without individual patient consent for medical research when it is not possible to use anonymised information and when seeking consent is not practical.

**Provenance and peer review** Not commissioned; externally peer reviewed.

**Data availability statement** Data may be obtained from a third party and are not publicly available. Due to data sharing agreements, we are not able to share these data. Researchers interested in using these data should consult the websites for the NCKDA (https://www.lshtm.ac.uk/research/centres-projects-groups/ckdaudit#welcome) and MINAP (https://www.nicor.org.uk/national-cardiac-audit-programme/myocardial-ischaemia-minap-heart-attack-audit/).

**ORCID iD**
Patrick Bidulka http://orcid.org/0000-0001-7644-2030

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
