## [Reviewer comments · BMJ Open]

ARTICLE DETAILS

TITLE (PROVISIONAL)	The impact of chronic kidney disease on case ascertainment for hospitalised acute myocardial infarction: An English cohort study
AUTHORS	Bidulka, Patrick; Scott, Jemima; Taylor, Dominic; Udayaraj, Udaya; Caskey, Fergus; Teece, Lucy; Sweeting, Michael; Deanfield, John; de Belder, Mark; Denaxas, S; Weston, Clive; Adlam, David; Nitsch, Dorothea

VERSION 1 – REVIEW

REVIEWER	Sood, Manish Ottawa Hospital Research Institute, Nephrology
REVIEW RETURNED	25-Oct-2021

GENERAL COMMENTS	Bidulka et al present a comprehensive, multi-objective case ascertainment examining AMI capture based on CKD severity and death and agreements between primary/secondary care of CKD severity. Using multiple retrospective administrative databases, they found variable capture from 11 to 66% depending on data source with differences based on CKD severity, higher capture if associated with death, and poor agreement between primary and secondary data sources for CKD severity. They conclude AMI case ascertainment is incomplete and suggest methods of improvement. overall a well done and important addition to the literature as data completion and validation studies are scant and required. Revisions/comments/questions: -consider shortening the study objectives in the abstract (maybe state "AMI case ascertainment with death, CKD and CKD severity")- consider adding an introductory or background statement on why this study is needed in the abstract-consider adding key definitions the abstract methods such as eGFR/Scr for CKD; CKD severity- how was AMI misclassification due to CKD handled? Were any additional criteria examined (ECG, troponin level; 2 diagnoses in records)?- would use established CKD nomenclature including CKD stage; see https://kdigo.org/conferences/nomenclature/- consider examining baseline CKD as per the KDIGO guidelines/definition as 2 measures > 90 days apart?-not sure you should have included those with no eGFR measure, even if at risk; I think more justification is required for this- I would exclude those on dialysis as they are a separate risk category altogether
--

	- a validation component (with chart review) would have greatly strengthened the study; are the investigators planning one in the future? -I would consider repeating the models with exclusions listed above as a sensitivity analyses - how was imputation handled? - a major limitation is the regional application of the findings as presented; how could this work be utilized by Non-UK readers?
--	---

REVIEWER	Sato , A Tsukuba Daigaku Igaku Bumon, Cardiology
REVIEW RETURNED	03-Dec-2021

GENERAL COMMENTS	The authors aimed to determine: (1) The impact of chronic kidney disease (CKD) severity on the completeness and validity of case ascertainment for hospitalised AMI between two secondary care datasets in England; (2) whether risk of death is associated with the dataset in which AMI is recorded; (3) agreement between primary and secondary care estimations of CKD severity. They concluded that case ascertainment for AMI hospitalisations is incomplete in both MINAP and HES and is associated with CKD severity. The author's manuscripts are actual and clinically relevant design protocol. However, several issues should be considered to assess the results in this paper. My comments are related to the following points: 1) The strength of this paper is a large sample size of AMI events. Previous studies demonstrated incomplete capture of AMI events and extend them to a population with CKD. What are the differences and highlights compared to some studies reported in the past? 2) Why generalizability to the general population is limited as NCKDA only included people with CKD and/or risk factors for CKD? How do the authors solve the issues of this incomplete capture of AMI events? Please discuss it more detail.
--

REVIEWER	Ali, Hatem University Hospitals Birmingham NHS Foundation Trust
REVIEW RETURNED	07-Dec-2021

GENERAL COMMENTS	I have read with great interest this important study titled: "The impact of chronic kidney disease on case ascertainment for hospitalised acute myocardial infarction: An English cohort study". The study assess the case ascertainment for AMI in the UK and its relation with CKD It found Case ascertainment for AMI hospitalisations is incomplete in both MINAP and HES and is associated with CKD I have the following comments: Abstract and introduction: Concise and well written. Abstract concise and to the point. Introduction clearly explains the aim and the importance of the study. Methodology and results: -Totally agree with categorising the Dialysis population into severe CKD or CKD4-5 category without considering the latest GFR.
--

	-In the secondary outcomes. You did:” We investigated the agreement between CKD stage derived from the most recent primary care SCr test (NCKDA data) and CKD stage derived from the secondary care SCr test conducted within 24 hours of AMI hospitalisation (MINAP data). “ I was not surprised to find poor correlation between CKD at the primary care and at hospitalization. However, my question is: How did you know that the GFR at time of hospitalization actually reflects a chronic kidney disease state and not an ACUTE KIDNEY INJURY? A rise in creatinine at the time of hospitalization for AMI could actually be more likely an AKI not a progression of CKD. This is due to Acute myocardial infarction (AMI) is one of the critical conditions that triggers AKI. This is in part due to the existence of comorbid factors, hemodynamic instability, and the use of medicines with kidney toxicity. Studies indicated that the incidence of AKI ranged from 7.1% to 29.3% during hospitalization in patients with AMI. Moreover, when AMI was complicated by cardiogenic shock, the occurrence of AKI reached more than 50%. Among patients with AMI, those with AKI had a 20- to 40-fold higher mortality rate in comparison to the one without AKI. -Was there any relation between mortality rates and the creatinine level at time of hospitalization ? Would the creatinine level at time of hospitalisation be a prognostic factor for mortality in AMI patients? Discussion: You said: “mortality. CKD stage calculated from SCr at the time of AMI admission (recorded in MINAP) varied significantly from that derived from SCr recorded in primary care. How do you know that the creatinine level at time of hospitalization is not Acute kidney injury secondary to the AMI?
--	---

VERSION 1 – AUTHOR RESPONSE

No	Reviewer’s comment	Author’s response	Location in revised manuscript (page)
Reviewer 1			
1.1	Bidulka et al present a comprehensive, multi-objective case ascertainment examining AMI capture based on CKD severity and death and agreements between primary/secondary care of CKD severity. Using multiple retrospective administrative databases, they found variable capture from 11 to 66% depending on data source with differences based on CKD severity, higher capture if associated with death, and poor agreement between primary and secondary data sources for CKD severity. They conclude AMI case	Thank you for your time reviewing this manuscript.	

	ascertainment is incomplete and suggest methods of improvement. overall a well done and important addition to the literature as data completion and validation studies are scant and required.		
1.2	-consider shortening the study objectives in the abstract (maybe state "AMI case ascertainment with death, CKD and CKD severity")	We have re-worded the objective section of the abstract to make this section more concise. This section now reads: "We aimed to determine the association between CKD severity and AMI case ascertainment in two secondary care datasets, and the agreement in estimated glomerular filtration rate (eGFR) between the same datasets."	4
1.3	- consider adding an introductory or background statement on why this study is needs in the abstract	We have re-worded the objective section of the abstract to provide additional background information justifying the purpose of this study. We agree with the reviewer this is important. The objectives section now begins: "Acute myocardial infarction (AMI) case ascertainment improves for the UK general population using linked health datasets. Because care pathways for people with chronic kidney disease (CKD) change based on disease severity, AMI case ascertainment for these people may differ."	4
1.4	-consider adding key definitions the abstract methods such as eGFR/Scr for CKD; CKD severity	Unfortunately, when keeping to the 300-word limit set by the journal, we cannot include these details in the abstract. However, we have written a 392-word version of the abstract, included at the end of the Main document (pages 35-36), which includes these details. We have asked the editorial team of the journal to advise which version is most appropriate. The explanation of the key CKD definitions in this longer version reads: "CKD status was defined using estimated glomerular filtration rate (eGFR), derived from the most recent serum creatinine value recorded in primary care. Moderate-severe CKD was defined as eGFR<60mL/min/1.73m ² , and mild CKD or at risk of CKD was defined as eGFR≥60mL/min/1.73m ² or eGFR missing. CKD stages were grouped as follows: (1) At risk of CKD & Stages 1-2 (eGFR missing or ≥60mL/min/1.73m ²), (2) Stage 3a (eGFR 45-59mL/min/1.73m ²), (3) Stage 3b (eGFR 30-44mL/min/1.73m ²), and (4) Stages 4-5 (eGFR<30mL/min/1.73m ²)."	300-word abstract: pages 4-5, 392-word abstract: 35-36
1.5	- how was AMI misclassification due to CKD handled? Were any additional criteria examined (ECG, troponin level; 2 diagnoses in records)?	To define AMI in the MINAP secondary care dataset, we used ECG results, troponin levels, and elevated cardiac markers to identify AMI using an algorithm previously developed by clinicians and epidemiologists. We briefly describe this algorithm in the methods section (page 12), and in more detail in the supplementary table 1. Unfortunately, these same data are not recorded in HES. Therefore, we used ICD-10 codes (I.21-23) to	26-27

		identify AMI in this dataset, which we also explain in the methods section (page 12). We made an addition to the limitations section of the discussion to mention that we lacked important data in HES, which were available in MINAP, that could have reduced AMI misclassification. This amended section now reads: “Finally, AMI misclassification in HES data may have occurred due to the structure and level of detail available in this dataset. For example, we may have missed AMI cases by including only those recorded in the first diagnostic position of the first episode of a HES admission; however, our sensitivity analysis which included AMI hospitalisations recorded in the first or second diagnostic position showed similar results. In addition, unlike MINAP data, HES data do not include ECG results and troponin levels, which we could have used to reduce potential misclassification.”	
1.6	- would use established CKD nomenclature including CKD stage; see https://kdigo.org/conferences/nomenclature/	Thank you for drawing our attention to this. We have removed our use of “eGFR/CKD category” throughout the document. In response to the comments made by Reviewer 3 (comments 3.4 and 3.6), we also now use the term “eGFR stage” to refer to the level of eGFR determined by using the serum creatinine value at the time of acute hospital admission. This is because the use of SCr taken during AMI hospitalisation has not previously been validated as an accurate means of determining baseline CKD stage. We have explained our reasoning in the exposure section of the methods: “As the use of a single serum creatinine test at the time of AMI hospitalisation to determine CKD stage has not previously been validated, we have used the term “eGFR stage” in place of CKD stage to refer to the eGFR level calculated from this test.”	11, 14, 15, 22, 25
1.7	- consider examining baseline CKD as per the KDIGO guidelines/definition as 2 measures > 90 days apart?	Thank you for this comment. We chose not to require two measures >90 days apart for our definition of baseline CKD since the population we are investigating is at high risk of mortality since everyone has experienced an AMI event. In addition, restricting to a baseline population having all two measurements of eGFR prior to an AMI hospitalisation would introduce a significant selection bias, as many patients with early CKD are only found to have CKD just before or during admission. There will be a subset of people who may not survive long enough to get two eGFR measures and will thus be excluded from the cohort. We recognise that we may misclassify some people as having CKD when in fact the low eGFR recorded in primary care may reflect acute illness (e.g. acute kidney injury), although we are confident this is less	26

		of an issue in primary vs secondary care. We have added this to the limitation section of the discussion. It now reads: “Furthermore, we included people with at least one reduced kidney function test as potentially having CKD since not every patient undergoes regular CKD testing in our routine clinical datasets. Defining CKD using one eGFR measurement will have led to some misclassification. However, as people with CKD have very high cardiovascular risk and because of the infrequent serum creatinine measurement in primary care, applying the chronicity criterion would have led to a selected cohort of people who did not develop a myocardial infarction until the second measurement had been done.”	
1.8	-not sure you should have included those with no eGFR measure, even if at risk; I think more justification is required for this	Thank you for your comment. We know from previous validation studies in UK primary care that people without eGFR measurements are much less likely to have CKD than those who do (Iwagami et al, 2017). We therefore chose to keep these people in our analysis. We agree further explanation is needed in the methods. We have added the following sentence and citation to our exposure definition (methods, page 10) to justify our inclusion of people with no creatinine measure recorded in primary care in our analysis: “We assumed people with no eGFR recorded in primary care did not have moderate to severe CKD since these people are much less likely to have CKD than those with eGFR recorded.²²” We have also included this as a potential limitation in the discussion section (page 26). This section reads: “We may have incorrectly misclassified people who have no documented tests for CKD in primary care as having risk factors for CKD only; however, previous work has shown this group of people are much less likely to have CKD than those who do have CKD tests recorded in primary care.²²” Reference ²²IWAGAMI, M., TOMLINSON, L. A., MANSFIELD, K. E., CASULA, A., CASKEY, F. J., AITKEN, G., FRASER, S. D. S., RODERICK, P. J. & NITSCH, D. 2017. Validity of estimated prevalence of decreased kidney function and renal replacement therapy from primary care electronic health records compared with national survey and registry data in the United Kingdom. Nephrol Dial Transplant, 32, ii142-ii150.	10, 26
1.9	- I would exclude those on dialysis as they are a separate risk category altogether	Thank you for your suggestion. We have chosen to add a sensitivity analysis where we drop those people who had a record of dialysis when calculating the associations between CKD status and AMI case ascertainment, and AMI case ascertainment and risk	Pages 15, 23, supplementary tables 10-11

		of death (in-hospital, and post-discharge). We agree these people are substantially different from the overall cohort and this sensitivity analysis is necessary. We did not observe any appreciable differences in our results after conducting this sensitivity analysis. We added the results in tables to the supplement (see Supplementary tables 10-11). We also described this sensitivity analysis in the methods (see page 15, under the sub-heading “Sensitivity analyses”) and in the results section (see page 23, under the sub-heading “Sensitivity analyses”).	
1.10	- a validation component (with chart review) would have greatly strengthened the study; are the investigators planning one in the future?	Unfortunately, these data are de-identified and we do not have ethical permission, nor the resources, to identify these data, obtain consent from individuals, and review chart histories. In addition, given the size of this cohort a chart review would not be feasible.	
1.11	-I would consider repeating the models with exclusions listed above as a sensitivity analyses	As the reviewer suggested, we have added a sensitivity analysis where we drop people who have a history of receiving dialysis from the analysis. Please see details in our response to Reviewer comment 1.9.	Pages 15, 23, supplementary tables 10-11
1.12	- how was imputation handled?	Thank you for this comment. We have made an edit to the “Missing data” section of the methods to specify that we used a complete case analysis when building our multivariable models. People with missing ethnicity (1%) and IMD data (<1%) were excluded prior to building our unadjusted, partially adjusted, and adjusted multinomial models. We used this approach instead of imputation since we do not think these data are missing at random, and imputation would therefore be invalid. Because missing data for variables included in our multivariable models were minimal, we expect our results to be minimally affected by excluding incomplete cases.	16
1.13	- a major limitation is the regional application of the findings as presented; how could this work be utilized by Non-UK readers?	We agree that these findings are most relevant to UK readers and researchers. However, we have edited the “Future research” section to invite further research in other settings similar to this study to improve the quality of observational research. This section now reads: “This study demonstrates how AMI case ascertainment in England can be improved by using linked healthcare datasets. Further research investigating cardiovascular and kidney disease incidence, prevalence and outcomes should follow this approach. Other countries with similarly rich, yet fragmented healthcare datasets would benefit from applying similar methods to evaluate the validity and completeness of cardiovascular and kidney disease capture in similar data. Optimising data quality in healthcare datasets and simplifying the process of data linkage would facilitate high-quality observational research to inform the design of future RCTs and provide estimated treatment effects where RCT data are lacking.”	27

Reviewer 2			
2.1	The authors aimed to determine: (1) The impact of chronic kidney disease (CKD) severity on the completeness and validity of case ascertainment for hospitalised AMI between two secondary care datasets in England; (2) whether risk of death is associated with the dataset in which AMI is recorded; (3) agreement between primary and secondary care estimations of CKD severity. They concluded that case ascertainment for AMI hospitalisations is incomplete in both MINAP and HES and is associated with CKD severity.	Thank you for your time reviewing this manuscript.	
2.2	The author's manuscripts are actual and clinically relevant design protocol. However, several issues should be considered to assess the results in this paper. My comments are related to the following points: 1) The strength of this paper is a large sample size of AMI events. Previous studies demonstrated incomplete capture of AMI events and extend them to a population with CKD. What are the differences and highlights compared to some studies reported in the past?	Thank you for your comment. We agree it is important to contextualise these findings with respect to previously published literature. In our discussion section on pages 24-25, we compare these findings to previously published papers which investigated AMI case ascertainment in the general population using the same datasets (HES and MINAP). We found that we had similar agreement between HES and MINAP AMI case ascertainment (42%) among people with moderate-severe CKD as in the general population investigated in the paper by Herret et al, 2013 (46%) and by Torabi et al, 2015 (32%). Our findings are dissimilar when looking at people with mild CKD or at-risk of CKD (~11%). We suggest possible explanations for this poor agreement among this particular subgroup on pages 24-25. Our finding that people with AMI recorded in only one dataset have a greater probability of death during AMI hospitalisation is similar to the results of Herret et al, 2013. We discuss this similarity on page 25. We hope the reviewer finds these comparisons relevant and informative in our discussion. References: Herrett E, Shah AD, Boggon R, et al. Completeness and diagnostic validity of recording acute myocardial infarction events in primary care, hospital care, disease registry, and national mortality records: cohort study. BMJ (Clinical research ed). 2013;346:f2350. Torabi A, Cleland JG, Sherwi N, et al. Influence of case definition on incidence and outcome of acute coronary syndromes. Open Heart. 2016;3(2):e000487. doi:10.1136/openhrt-2016-000487	
2.3	2) Why generalizability to the general population is limited as NCKDA only	Thank you for your comment. The NCKDA was designed to audit care of people at risk of CKD or	26

	included people with CKD and/or risk factors for CKD? How do the authors solve the issues of this incomplete capture of AMI events? Please discuss it more detail.	with CKD in primary care in England and Wales. Relevant data were extracted from the primary care records in 2 main cross-sectional extracts. Data from NCKDA participants were linked with HES and MINAP to define our study cohort. Due to the nature of the NCKDA audit, we cannot include people who were not at risk of developing CKD or had CKD during the audit period. Therefore, we cannot directly solve the problem of poor generalisability outside the population at risk of kidney disease. However, given that our study question focuses on any differences in AMI capture among people with different levels of kidney disease, we do not believe this is a problem. We have tried to contextualise our findings by comparing our results with other studies which studied the general population. We found important differences between AMI case ascertainment in our study population (people at risk of CKD or with CKD) versus other study populations (the general population). These differences are highlighted on pages 24-25 of the discussion, with possible explanations for these differences. In the limitations section of the discussion, we acknowledge our inability to directly compare AMI case ascertainment between people with poor kidney function and the general population without risk factors for CKD (page 26). We have edited this sentence to make it clearer we cannot generalise to people without risk factors for CKD, since those with risk factors for CKD would have been included in our cohort. This sentence now reads: “The NCKDA only included people with CKD and/or risk factors for CKD; therefore, we cannot generalise our results to people without risk factors for CKD.” We hope the reviewer finds this explanation satisfactory. We are happy to respond to further questions.	
Reviewer 3			
3.1	I have read with great interest this important study titled: “The impact of chronic kidney disease on case ascertainment for hospitalised acute myocardial infarction: An English cohort study”. The study assess the case ascertainment for AMI in the UK and its relation with CKD It found Case ascertainment for AMI hospitalisations is incomplete in both MINAP and HES and is associated with CKD	Thank you for your time reviewing this manuscript.	
3.2	I have the following comments: Abstract and introduction:	Thank you for this comment. We have made edits to our abstract in response to comments from Reviewer	4-5

	Concise and well written. Abstract concise and to the point. Introduction clearly explains the aim and the importance of the study.	1. We hope the reviewer finds these edits are appropriate.	
3.3	Methodology and results: -Totally agree with categorising the Dialysis population into severe CKD or CKD4-5 category without considering the latest GFR.	Thank you for your comment.	
3.4	-In the secondary outcomes. You did.” We investigated the agreement between CKD stage derived from the most recent primary care SCr test (NCKDA data) and CKD stage derived from the secondary care SCr test conducted within 24 hours of AMI hospitalisation (MINAP data). “ I was not surprised to find poor correlation between CKD at the primary care and at hospitalization. However, my question is: How did you know that the GFR at time of hospitalization actually reflects a chronic kidney disease state and not an ACUTE KIDNEY INJURY? A rise in creatinine at the time of hospitalization for AMI could actually be more likely an AKI not a progression of CKD. This is due to Acute myocardial infarction (AMI) is one of the critical conditions that triggers AKI. This is in part due to the existence of comorbid factors, hemodynamic instability, and the use of medicines with kidney toxicity. Studies indicated that the incidence of AKI ranged from 7.1% to 29.3% during hospitalization in patients with AMI. Moreover, when AMI was complicated by cardiogenic shock, the occurrence of AKI reached more than 50%. Among patients with AMI, those with AKI had a 20- to 40-fold higher mortality rate in comparison to the one without AKI.	Thank you for this comment. We agree that people with poor kidney function admitted to hospital for AMI are likely to develop co-incident AKI, and that this is associated with worse prognosis. Thus, a low eGFR at the time of AMI hospitalisation does not necessarily indicate chronic kidney disease; rather, it could be AKI that develops alongside AMI. As we explain in the introduction, this justifies our analyses investigating the agreement between eGFR from primary care and eGFR measured within 24 hours of AMI hospitalisation since several studies we cited have relied on secondary care data to identify study populations with CKD and define baseline CKD stage. To be clear that we, like the reviewer, are sceptical that eGFR at the time of AMI hospitalisation is a good estimate of baseline CKD stage, we have edited the terminology we use to describe objective 3 analyses. The objective now reads: “Objective 3 – Agreement between eGFR in primary and secondary care” rather than “Objective 3 – Agreement between CKD staging” (see page 14, methods section. We have edited the manuscript to use the term “eGFR stage” instead of “CKD stage” throughout our description of methods and results when describing eGFR categorisation using the eGFR at AMI hospitalisation (pages 12-15, 22). When designing this study, we considered that eGFR at AMI admission could be systematically lower than the most recent eGFR in primary care (suggesting AKI). However, the data are not consistent with this hypothesis - we did not observe the eGFR at AMI admission to either under or over-estimate baseline eGFR status (generated from primary care data). We hypothesise that since changes in serum creatinine are unlikely to show within 24 hours of AMI onset, we do not see the systematic under-estimation of kidney function when using eGFR within 24 hours of AMI admission. Rather, we see variation by about 30% around the mean, which is as expected when using the MDRD equation to calculate eGFR.^{23,24} We have added a statement to the first paragraph of the discussion to emphasise our finding of no systematic under- or over-estimate of eGFR when	12-15, 22, 24, Table 4

		using eGFR at AMI hospitalisation. The statement reads: “There was marked variation between eGFR at AMI admission and preceding eGFR measurements in primary care, but no obvious systematic bias in terms of over/underestimation of eGFR at AMI admission.” References 23. Levey AS, Coresh J, Greene T, et al. Using standardized serum creatinine values in the modification of diet in renal disease study equation for estimating glomerular filtration rate. Ann Intern Med. Aug 15 2006;145(4):247-54. doi:10.7326/0003-4819-145-4-200608150-00004 24. Levey AS, Stevens LA, Schmid CH, et al. A new equation to estimate glomerular filtration rate. Ann Intern Med. May 5 2009;150(9):604-12. doi:10.7326/0003-4819-150-9-200905050-00006	
3.5	-Was there any relation between mortality rates and the creatinine level at time of hospitalization ? Would the creatinine level at time of hospitalisation be a prognostic factor for mortality in AMI patients?	Thank you for this comment. We consider this to be an important and significant question. We are currently drafting a second paper using the same cohort, which will investigate processes of care (e.g. angiography, PCI, CABG) and outcomes (e.g. mortality, re-admissions) with respect to baseline CKD status and eGFR category at AMI hospitalisation.	
3.6	Discussion: You said: “mortality. CKD stage calculated from SCr at the time of AMI admission (recorded in MINAP) varied significantly from that derived from SCr recorded in primary care. How do you know that the creatinine level at time of hospitalization is not Acute kidney injury secondary to the AMI?	Thank you for this comment. We agree that we cannot know if the difference in creatinine level at time of hospitalisation is due to AKI co-incident with the AMI, or any other explanation. We proposed possible reasons to explain the differences found between the secondary care and primary care eGFRs on pages 25-26. These reasons include AKI which develops along with AMI, as well as other possible explanations such as known variability of eGFR when using the MDRD formula,^{23,24} progression of CKD, differential use of medication that affects the renin-angiotensin-aldosterone system, and variation around the mean (page 26). Since we did not observe a systematic decrease in eGFR at the time of AMI admission compared with eGFR in primary care (which would suggest AKI), we could not limit our possible explanations to only AKI. We hope the reviewer agrees this is a sensible approach. We edited the discussion concerning these results, which now reads: “Across all levels of eGFR, we found significant variation between eGFR stage derived from SCr	24-26

taken within 24 hours of AMI admission (recorded in MINAP) and that derived from SCr in primary care, which is in line with reported variability of eGFR in validation studies.^{23,24} As expected with known limitations of using MDRD eGFR to estimate kidney function for GFRs above 60mL/min/1.73m², binary classification between individuals with CKD stages 3-5 and those with stages 1-2 is more reliable than classification by CKD stage. These findings suggest that although previous research¹³⁻¹⁵ using SCr at AMI admission recorded in MINAP as a proxy for baseline CKD stage may result in misclassification, it is unlikely to have resulted in a systematic bias in either over- or under-estimation of CKD stage. Differences between SCr recorded in primary care and SCr recorded in MINAP may reflect progression of CKD, differential use of medication that affects the renin-angiotension-aldosterone system, AKI at the time of serum sampling (although changes in serum creatinine are unlikely to show within 24 hours of AMI onset), known variability of eGFR when using the MDRD formula, or variation around the mean.”

References

13. Gupta T, Paul N, Kolte D, et al. Association of chronic renal insufficiency with in-hospital outcomes after percutaneous coronary intervention. *J Am Heart Assoc.* Jun 16 2015;4(6):e002069. doi:10.1161/jaha.115.002069
14. Rozenbaum Z, Benchetrit S, Minha S, et al. The Effect of Admission Renal Function on the Treatment and Outcome of Patients with Acute Coronary Syndrome. *Cardiorenal Med.* Jun 2017;7(3):169-178. doi:10.1159/000455239
15. Shaw C, Nitsch D, Steenkamp R, et al. Inpatient coronary angiography and revascularisation following non-ST-elevation acute coronary syndrome in patients with renal impairment: a cohort study using the Myocardial Ischaemia National Audit Project. *PLoS One.* 2014;9(6):e99925. doi:10.1371/journal.pone.0099925
23. Levey AS, Coresh J, Greene T, et al. Using standardized serum creatinine values in the modification of diet in renal disease study equation for estimating glomerular filtration rate. *Ann Intern Med.* Aug 15 2006;145(4):247-54. doi:10.7326/0003-4819-145-4-200608150-00004
24. Levey AS, Stevens LA, Schmid CH, et al. A new equation to estimate glomerular filtration rate. *Ann Intern Med.* May 5 2009;150(9):604-12. doi:10.7326/0003-4819-150-9-200905050-00006

VERSION 2 – REVIEW

REVIEWER	Sood, Manish Ottawa Hospital Research Institute, Nephrology
REVIEW RETURNED	17-Jan-2022

GENERAL COMMENTS	the investigators did a good job defending their work and responding to revisions
---

REVIEWER	Sato , A Tsukuba Daigaku Igaku Bumon, Cardiology
REVIEW RETURNED	01-Feb-2022

GENERAL COMMENTS	There are no comments for your revision. Thank you for the change and the explanations.
---

REVIEWER	Ali, Hatem University Hospitals Birmingham NHS Foundation Trust
REVIEW RETURNED	25-Jan-2022

GENERAL COMMENTS	The authors have answered my queries satisfactorily. I have no further comments I would only suggest to explain in the discussion their approach about dealing with the possibility of having AKI at the time of admission , as explained to me in the response to reviewer
--

VERSION 2 – AUTHOR RESPONSE

No	Reviewer's comment	Author's response	Location in revised manuscript (page)
Reviewer 1			
	the investigators did a good job defending their work and responding to revisions	Thanks again for taking the time to review our manuscript.	
Reviewer 2			
	There are no comments for your revision. Thank you for the change and the explanations.	Thanks again for taking the time to review our manuscript.	
Reviewer 3			
	The authors have answered my queries satisfactorily. I have no further comments I would only suggest to explain in the discussion their approach about dealing	Thanks again for taking the time to review our manuscript.	26

with the possibility of having AKI at the time of admission , as explained to me in the response to reviewer

We have made an addition to the text of our discussion which makes clear that our original hypothesis was that we would observe a systematic under-estimation of kidney function at the time of AMI hospitalisation since it is well known that risk of AKI is substantially increased during an AMI episode, as we explained to the reviewer in the previous response letter. The section now reads (with additions highlighted in yellow):

As expected with known limitations of using MDRD eGFR to estimate kidney function for GFRs above 60mL/min/1.73m², binary classification between individuals with CKD stages 3-5 and those with stages 1-2 is more reliable than classification by CKD stage. These findings suggest that although previous research¹³⁻¹⁵ using SCr at AMI admission recorded in MINAP as a proxy for baseline CKD stage may result in misclassification, it is unlikely to have resulted in a systematic bias in either over- or under-estimation of CKD stage, despite our initial hypothesis that there would be systematic underestimation of kidney function due to the substantially increased risk of AKI during an AMI hospitalisation.³¹ Differences between SCr recorded in primary care and SCr recorded in MINAP may reflect progression of CKD, differential use of medication that affects the renin-angiotension-aldosterone system, AKI at the time of serum sampling (although changes in serum creatinine are unlikely to show within 24 hours of AMI onset), or variation around the mean.

31. Pickering JW, Blunt IRH, Than MP. Acute Kidney Injury and mortality prognosis in Acute Coronary Syndrome patients: A meta-analysis. *Nephrology*. 2018;23(3):237-246. doi:<https://doi.org/10.1111/nep.12984>

		We hope this addition is appropriate and the reviewer is happy with the updated version of the manuscript.	
--	--	--	--